# Post-Diagnosis Vitamin D Supplement Use and Survival among Cancer Patients: A Meta-Analysis

**DOI:** 10.3390/nu14163418

**Published:** 2022-08-19

**Authors:** Qiao-Yi Chen, Sohyun Kim, Bohyoon Lee, Gyeongin Jeong, Dong Hoon Lee, NaNa Keum, JoAnn E. Manson, Edward L. Giovannucci

**Affiliations:** 1Department of Food Science and Biotechnology, Dongguk University, Goyang 10326, Korea; 2Department of Nutrition, Harvard T.H. Chan School of Public Health, Boston, MA 02115, USA; 3Department of Medicine, Brigham and Women’s Hospital, Harvard Medical School, Boston, MA 02115, USA; 4Department of Epidemiology, Harvard T.H. Chan School of Public Health, Boston, MA 02115, USA

**Keywords:** vitamin D supplement use, post-diagnosis, overall survival, progression-free survival, cancer-specific survival, relapse, meta-analysis, randomized controlled trial, cohort study

## Abstract

Vitamin D administered pre-diagnostically has been shown to reduce mortality. Emerging evidence suggests a role of post-diagnosis vitamin D supplement intake for survival among cancer patients. Thus, we conducted a meta-analysis to evaluate the relationship. PubMed and Embase were searched for relevant observational cohort studies and randomized trials published through April 2022. Summary relative risk (SRR) and 95% confidence interval (CI) were estimated using the DerSimonian–Laird random-effects model. The SRR for post-diagnosis vitamin D supplement use vs. non-use, pooling cohort studies and randomized trials, was 0.87 (95% CI, 0.78–0.98; *p* = 0.02; *I*^2^ = 0%) for overall survival, 0.81 (95% CI, 0.62–1.06; *p* = 0.12; *I*^2^ = 51%) for progression-free survival, 0.86 (95% CI, 0.72–1.03; *p* = 0.10; *I*^2^ = 0%) for cancer-specific survival, and 0.86 (95% CI, 0.64–1.14; *p* = 0.29; *I*^2^ = 0%) for relapse. Albeit not significantly heterogeneous by variables tested, a significant inverse association was limited to cohort studies and supplement use during cancer treatment for overall survival, and to studies with ≤3 years of follow-up for progression-free survival. Post-diagnosis vitamin D supplement use was associated with improved overall survival, but not progression-free or cancer-specific survival or relapse. Our findings require confirmation, as randomized trial evidence was insufficient to establish cause-and-effect relationships.

## 1. Introduction

Cancer is a global public health concern and the leading cause of death in most developed countries [1]. According to the International Agency for Research on Cancer, in 2020, there were approximately 19.3 million incident cancer cases and 10.0 million cancer deaths worldwide [2]. While early detection by screening programs and advances in treatment have improved cancer survival rates [2], dietary supplement use along with lifestyle modifications is of great interest for cancer patients to further increase their survival chance. Recent studies reported that over half of cancer patients initiated dietary supplement use following diagnosis [3], including a wide variety of vitamins, minerals, herbal supplements, and nutraceuticals (e.g., ginseng) [3]. Notably, vitamin D is one of the most widely used dietary supplements, because it has been hypothesized to decrease adverse symptoms during cancer treatments, to reduce cancer recurrence, to boost the immune system and energy level, among other benefits [4,5].

Accumulating evidence suggests that vitamin D supplements may confer protective effects against total cancer mortality in “usual risk” populations [6]. In a recent meta-analysis of randomized controlled trials (RCTs), daily vitamin D supplementation was significantly associated with an approximately 13% reduced total cancer mortality [7]. Among cancer patients, a meta-analysis of three observational studies and two RCTs showed that post-diagnosis vitamin D supplement use was significantly associated with a 14% reduced all-cause mortality [3]. Since then, additional studies on vitamin D supplementation among cancer patients and survival have been published, suggesting that an update of the previous meta-analysis and an exploration of heterogeneity in the relationship could be informative. Therefore, we conducted a meta-analysis of post-diagnosis vitamin D supplement use among cancer patients and survival outcomes, while exploring heterogeneity by diverse factors, including the timing of vitamin D supplement use, the cancer type, the follow-up period, and the circulating level of 25-hydroxyvitamin D (25(OH)D).

## 2. Methods

This meta-analysis was performed and reported in accordance with the PRISMA guideline (Appendix A) [8].

### 2.1. Study Search

PubMed and Embase were searched for relevant articles through April 2022. The search was limited to human studies and the English language, but no other restrictions were imposed. The detailed search terms are provided in Appendix A.

### 2.2. Study Selection

To be included in this meta-analysis, studies had to be a RCT or cohort study that examined the relationship between vitamin D supplement intake after cancer diagnosis and survival outcomes (overall survival, progression-free survival, cancer-specific survival, or relapse). Abstracts, unpublished results, and review articles were excluded. When there were multiple publications from the same trial [9,10], we selected the publication with larger numbers of participants [9]. To identify additional papers, we also checked the reference lists of selected articles and previous systematic reviews. After a series of screening procedures, a total of 11 studies (5 RCTs [9,11,12,13,14], 6 cohort studies [15,16,17,18,19,20]) were eligible for this meta-analysis. This study selection process is summarized in Figure 1.

### 2.3. Data Abstraction

From each article, the following information was extracted: name of the first author, publication year, study design, definitions (i.e., type, timing, period) of vitamin D supplement use and control, cancer type, outcome definition (i.e., overall survival, progression-free survival, cancer-specific survival, and relapse), relative risk (RR), 95% confidence interval (CI), confounding factors adjusted for, and important characteristics of the study population (e.g., country, age, sex), and exclusion criteria regarding vitamin D supplements intake (Appendix A).

Two authors (QYC and SK) participated in the study search, study selection, and data abstraction independently, and any disagreement was resolved through discussion with NK.

### 2.4. Statistical Analyses

The summary RR (SRR) and 95% CI were calculated using the DerSimonian–Laird random-effects model for the association between post-diagnosis vitamin D supplement use and survival outcomes [21]. The *I*^2^ statistic was calculated to assess the degree of heterogeneity in the relationship across the studies [22]. Potential for small study effects, such as publication bias, were investigated by running Egger’s test [23]. Of note, one study [9], whose control group received 400 IU of vitamin D supplement rather than no supplement, reported the results based on one-sided statistical tests. By estimating the standard errors, we recalculated 95% CI based on two-sided tests and performed a sensitivity analysis excluding the study.

Potential sources of heterogeneity in the relationship were explored by performing subgroup meta-analyses and meta-regression according to a priori selected variables related to methodological characteristics (study design, follow-up period), etiologic heterogeneity (cancer type), and potential effect modifiers (timing of vitamin D supplement use relative to cancer treatment:during treatment vs. after treatment, baseline level of circulating 25(OH)D: <20 ng/mL vs. ≥20 ng/mL). Notably, because results of a meta-analysis of a small number of studies are not informative, subgroup analyses were not conducted when the total number of studies was less than five.

All the statistical tests were two-sided and *p* values of <0.05 were considered statistically significant. Analyses were performed using STATA 17 (StataCorp., College Station, TX, USA).

## 3. Results

### 3.1. Description of Included Studies

After screening 1742 publications, a total of 11 publications (5 RCTs [9,11,12,13,14], 6 observational cohort studies [15,16,17,18,19,20]) contributed to the meta-analyses for overall survival (9 studies), progression-free survival (8 studies), cancer-specific survival (3 studies), and relapse (3 studies). The main characteristics of the included studies are summarized in Appendix A.

### 3.2. Vitamin D Supplement Use after Cancer Diagnosis and Overall Survival

Based on a total of nine studies (three RCTs [11,12,14] and six cohort studies [15,16,17,18,19,20]), the SRR of overall survival associated with post-diagnosis vitamin D supplement use was 0.87 (95% CI, 0.78–0.98; *p* = 0.02), with no evidence of heterogeneity (*I*^2^ = 0%) (Figure 2A). Small study effects, such as publication bias, were not observed (*P_Egger_* = 0.83).

In subgroup analyses, there was no evidence of significant heterogeneity by the study design (*P_heterogeneity_* = 0.77), the cancer type (*P_heterogeneity_*, NA), the timing of vitamin D supplement use (*P_heterogeneity_* = 0.20), and the follow-up period (*P_heterogeneity_ =* 0.73) (Figure 3A–D). Nevertheless, a significant inverse association was observed among cohort studies (RR, 0.87; 95% CI, 0.77–0.98; *p* = 0.02; *I*^2^ = 0%) but not among RCTs (RR, 0.93; 95% CI, 0.62–1.38; *p* = 0.71; *I*^2^ = 0%) (Figure 3A); it was also observed when the supplement was used during treatment (RR, 0.78; 95% CI, 0.61–0.998; *p* = 0.048; *I*^2^ = 32%; three RCTs and four cohort studies), but not when the supplement was used after treatment (Figure 3C). There was a marginally significant inverse association in relation to breast cancer (RR, 0.88; 95% CI, 0.77–1.001; *p* = 0.05; *I*^2^ = 0%; five cohort studies), but not with other cancer types (Figure 3B).

### 3.3. Vitamin D Supplement Use after Cancer Diagnosis and Progression-Free Survival

Based on a total of eight studies (five RCTs [9,11,12,13,14] and three cohort studies [15,19,20]), the SRR of progression-free survival for vitamin D supplement use vs. non-use after cancer diagnosis was 0.81 (95% CI, 0.62–1.06; *p* = 0.12), with moderate heterogeneity (*I*^2^ = 51%) (Figure 2B). There was no evidence of small study effects such as publication bias (*P_Egger_* = 0.78). The aforementioned results remained unchanged in the sensitivity analysis, excluding a study [9] that tested high doses of vitamin D (8000/4000 IU/d) against the standard dose (400 IU/d) and reported the results based on a one-sided test (Data not shown).

The relationship between post-diagnosis vitamin D supplement use and progression-free survival was not significantly heterogeneous by the study design (*P_heterogeneity_* = 0.63), the cancer type (*P_heterogeneity_* = 0.75), the timing of vitamin D supplement use (*P_heterogeneity_* = 0.59), the follow-up period (*P_heterogeneity_* = 0.07), or the baseline level of circulating 25(OH)D (*P_heterogeneity_* = 0.61) (Figure 4A–E). Nevertheless, a significant inverse association was observed among studies with ≤3 years follow-up (RR, 0.61; 95% CI, 0.46–0.81; *p* = 0.001; *I*^2^ = 0%; two RCTs and two cohort studies), but not among studies > 3 years follow-up (Figure 4D).

### 3.4. Vitamin D Supplement Use after Cancer Diagnosis and Cancer-Specific Survival

Based on three studies (one RCTs [14] and two cohort studies [17,18]), the SRR of cancer-specific survival was 0.86 (95% CI, 0.72–1.03; *p* = 0.10) with no evidence of heterogeneity (*I*^2^ = 0%) (Figure 2C). Small study effects, such as publication bias, were not demonstrated (*P_Egger_* = 0.22). Heterogeneity in the relationship was not explored due to the small number of studies.

### 3.5. Vitamin D Supplement use after Cancer Diagnosis and Relapse

Based on three studies (two RCTs [13,14] and one cohort studies [18]), the SRR of relapse was 0.86 (95% CI, 0.64–1.14, *p* = 0.29) with no evidence of heterogeneity (*I*^2^ = 0%) (Figure 2D). Small study effects, such as publication bias, were not demonstrated (*P_Egger_* = 0.66). Due to the small number of studies, heterogeneity in the relationship was not explored for relapse.

## 4. Discussion

In this meta-analysis of observational cohort studies and RCTs, post-diagnosis vitamin D supplement use compared to non-use was associated with an approximately 13% improved overall survival, but was not associated with progression-free survival, cancer-specific survival, and relapse. The relationships did not vary significantly by the study design, the cancer type, the timing of vitamin D supplement use, and the follow-up period. Nevertheless, a significant inverse association with overall survival was limited to cohort studies and studies that assessed supplement use during treatment, and was suggestive for breast cancer. In relation to progression-free survival, a significant inverse association emerged among studies with ≤3 years of follow-up, which were the majority of the studies.

Despite a heightened interest in dietary supplements among cancer patients [24], previous studies on vitamin D supplement use and cancer outcomes were mostly conducted among individuals without cancer [7]. In a meta-analysis of RCTs that enrolled mostly individuals without cancer, daily vitamin D supplementation did not reduce total cancer incidence, but it was significantly associated with a 13% reduction in cancer mortality and 7% reduction in total mortality [7]. Carcinogenesis is a complex multiple process involving initiation, promotion, progression, and metastasis [25], and the divergent results between cancer incidence and cancer mortality suggest that vitamin D may operate at three potential stages: a pre-diagnostic period by influencing late-stage tumor progression and metastatic seeding, a during treatment period by interacting with cancer therapies, and a post-diagnostic period by improving survival [6].

In light of our current study, which evaluated studies on the association between vitamin D supplement use after cancer diagnosis and outcomes, finding null associations with most of the endpoints, the benefit of vitamin D supplements may be occurring primarily during the pre-diagnostic period by decreasing tumor invasiveness and metastatic potential. Nevertheless, we cannot completely rule out the potential benefits of vitamin D supplement during the post-diagnosis period. Firstly, we observed a significant inverse association between post-diagnosis vitamin D use and overall survival. Beneficial effects of vitamin D, such as immune-boosting, anti-inflammatory and insulin-sensitizing effects against non-cancer outcomes, may still operate after cancer diagnosis [26,27,28]. Secondly, in our subgroup analysis by baseline level of circulating 25(OH)D, an inverse association observed with progression-free survival, albeit not significant, was stronger for individuals with 25(OH)D ≥ 20 ng/mL (rather than <20 ng/mL). This suggests a potential threshold effect by which a large amount of vitamin D needs to be accumulated in the body first for a survival benefit to manifest in response to an increased intake of vitamin D supplements [29]. Indeed, a meta-analysis showed that cancer patients with higher levels of circulating 25(OH)D at or near the time of diagnosis had better survival [30]. Finally, given our findings that inverse associations with overall survival or progression-free survival were more apparent in studies assessing vitamin D supplements during cancer treatment (rather than after treatment) and in studies with a short-term follow-up period, vitamin D may complement or enhance the effects of cancer therapies.

By cancer type, evidence for a beneficial effect of post-diagnosis vitamin D supplement use was most suggestive for overall survival among breast cancer patients. This result may be in part attributable to methodological factors, because all studies included for this breast cancer meta-analysis were observational cohort studies, and active research in the field of breast cancer survivorship resulted in a large number of publications. Yet there are potential biological mechanisms, albeit not exclusively applicable to breast cancer, that might explain a role of vitamin D supplementation for breast cancer survivorship. Vitamin D receptor (VDR) is a nuclear receptor, which upon activation by 1,25-dihydroxyvitamin D inhibits cell proliferation and induces cell differentiation and apoptosis [31]. The VDR is expressed in breast cells, and a meta-analysis found that higher levels of VDR in the nucleus and cytoplasm of breast cancer cells were significantly associated with a better overall survival among breast cancer patients [31]. The anti-cancer effect of vitamin D is also exerted by suppressing the proliferative effect of estrogen, because vitamin D lowers estrogen levels by inhibiting the activity of aromatase, an enzyme that converts androgens into estrogens through aromatization [32]. Indeed, in a cohort study, post-diagnosis vitamin D supplement use was associated with a 36% decreased recurrence of estrogen receptor positive breast cancer, but not associated with the recurrence of estrogen receptor negative breast cancer [18].

Our study is the first meta-analysis that analyzed the association between post-diagnosis vitamin D supplement intake and cancer outcomes, addressing important heterogeneity in the relationship by the timing of vitamin D supplement use, the cancer type, the baseline 25(OH)D levels, and others. Yet, there are several limitations to acknowledge. First, as an inherent limitation of the meta-analysis, any biases present within individual studies, which cannot be corrected by the meta-analysis, may compromise the validity of our results. In fact, due to the limited number of studies available, our meta-analyses pooled RCTs and observational cohort studies together, except for subgroup analysis by study design. While the relationships were not significantly heterogeneous by study design, causal interpretation of our findings should be avoided. Second, because all studies compared supplement use vs. non-use except for one study (high vs. standard dose) [9], our study was unable to examine the dose–response relationship. Yet, accumulating evidence suggests that higher doses of vitamin D supplementation than the recommended dietary allowances (600–800 IU for adults) might be needed for vitamin D to exert an anti-cancer effect [33]. All of the RCTs included in our meta-analysis, despite varying dose and frequency, provided vitamin D supplements of ≤2000 IU/day except one study [9], which provided 8000 IU/day for cycle 1 followed by 4000 IU/day for subsequent cycles, and observed a significant inverse association with progression-free survival. Nevertheless, given conflicting evidence on the safety and efficacy of non-daily mega-dose vitamin D supplementation [34,35], more studies testing doses within the safety upper limit (<4000 IU/day) [36] are warranted to identify the optimal dose of vitamin D supplements that can exert anti-cancer benefits. Third, as most of the included studies were conducted among breast cancer or colorectal cancer patients, our results might not apply to other cancer sites.

In conclusion, post-diagnosis vitamin D supplement use among cancer patients was associated with an improved overall survival, but was not related to progression-free survival, cancer-specific survival, or relapse. RCT evidence was limited and insufficient to establish cause-and-effect relationships. Future studies are needed to confirm our findings as well as to identify the optimal supplementation strategy (e.g., dose and timing) and the subgroup of people who will benefit most from the supplementation.

## Figures and Tables

**Figure 1 nutrients-14-03418-f001:**
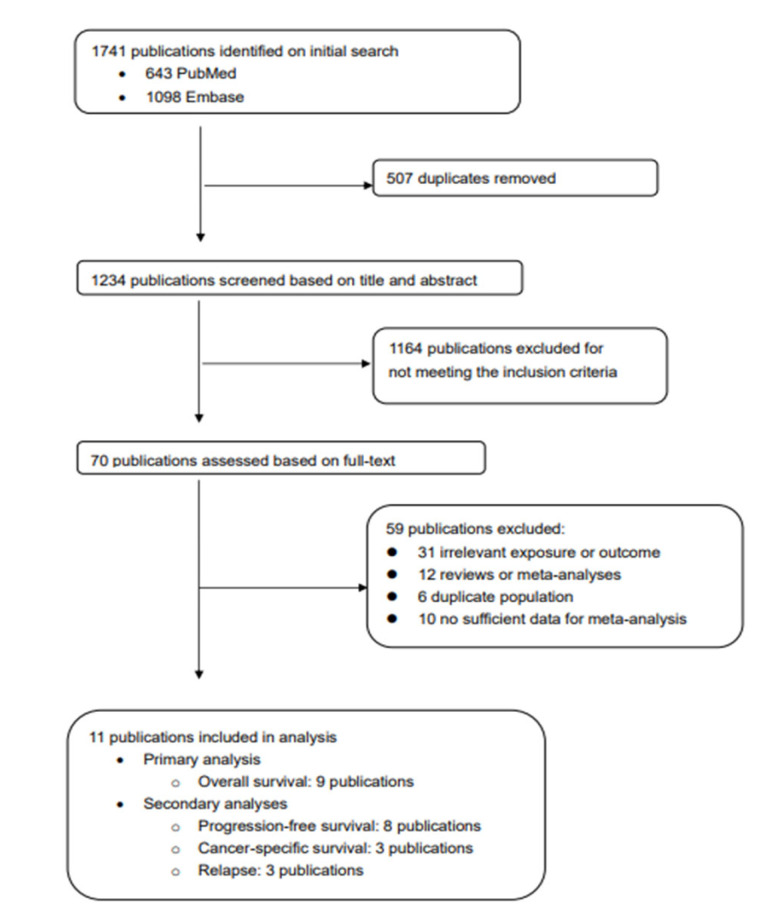
Flowchart for study selection.

**Figure 2 nutrients-14-03418-f002:**
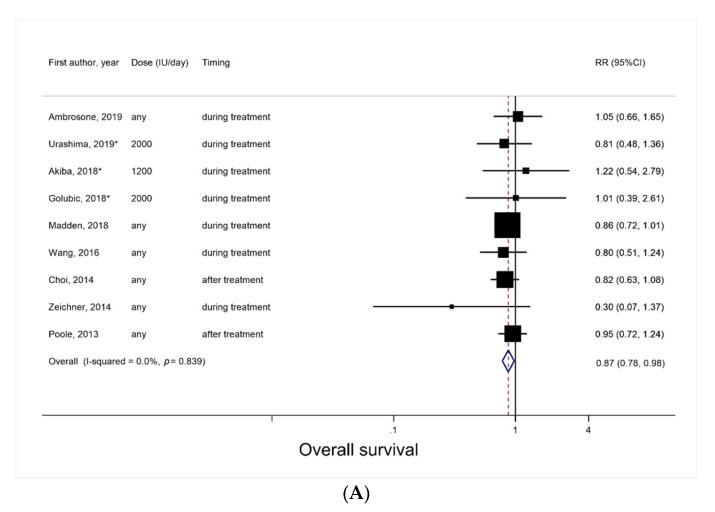
Forest plots for meta-analyses of post-diagnosis vitamin D supplement use and survival outcomes. (**A**) Overall survival; (**B**) Progression-free survival; (**C**) Cancer-specific survival; (**D**) Relapse. RCTs are marked with *. #: 100,000 IU/50 day, converted to 2000 IU/day.

**Figure 3 nutrients-14-03418-f003:**
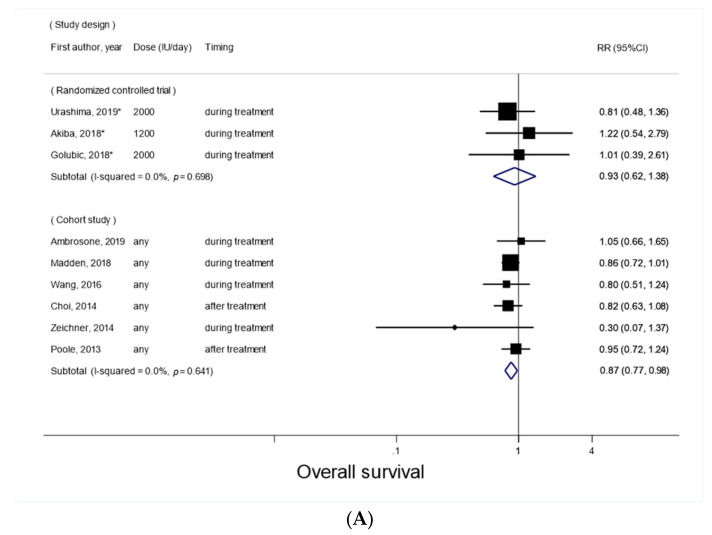
Forest plots for subgroup meta-analyses of post-diagnosis vitamin D supplement use and overall survival. (**A**) By study design; (**B**) By cancer type; (**C**) By timing of vitamin D supplement use; (**D**) By follow-up period. RCTs are marked with *.

**Figure 4 nutrients-14-03418-f004:**
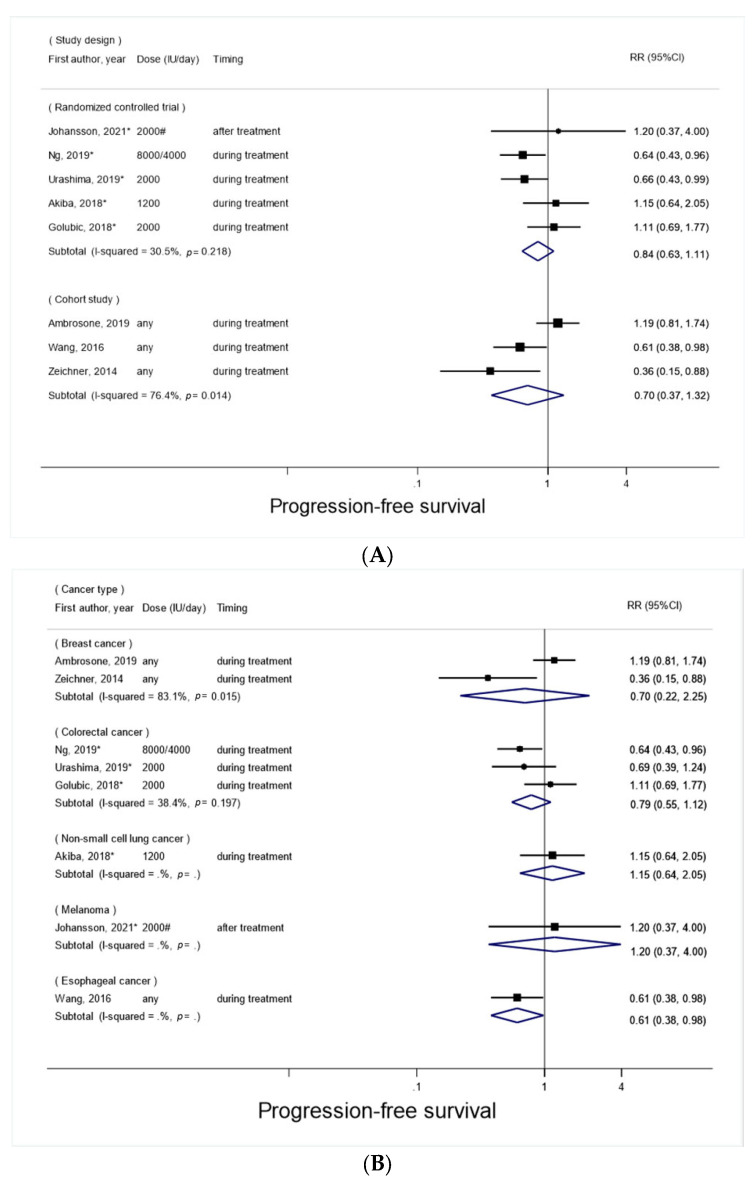
Forest plots for subgroup meta-analyses of post-diagnosis vitamin D supplement use and progression-free survival. (**A**) By study design; (**B**) By cancer type; (**C**) By timing of vitamin D supplement use; (**D**) By follow-up period; (**E**) By baseline level of circulating 25(OH)D. RCTs are marked with *. #: 100,000 IU/50 day, converted to 2000 IU/day.

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
