# Peer review of "Post-Diagnosis Vitamin D Supplement Use and Survival among Cancer Patients: A Meta-Analysis"

_nutrients, 2022, doi:10.3390/nu14163418_

Round 1
Reviewer 1 Report
The article is relevant, presents scientific novelty and is well designed. The results are clearly presented, and the conclusions supported by them. There are only a few suggestions:
1) It could be useful (for the readers) to include in the graphs regarding the timing of vitamin D supplement use, the respective dosage;
2) in the discussion the authors address the potential mechanisms that may explain the protection of vitamin D associated specifically with breast cancer. And what mechanisms might explain the lower overall mortality observed?
3) The discussion mentions that "an inverse association observed with progression-free survival, albeit not significant, was stronger for studies with higher 25(OH)D levels at baseline". It should be clarified in the discussion, for example, if in the analyzed studies the baseline values were indicative of vitamin D deficit or deficiency, or if, on the contrary, the participants had plasma concentrations of vitamin D within the reference values.
4) In conclusion it would be important to point out directions for clinical practice: based on the best current scientific evidence does it make sense to supplement all cancer patients with vitamin D? Does supplementation only make sense when changes (low concentrations) in plasma vitamin D concentration at diagnosis are observed? What dose(s) seem to be recommendable (and safe) to reduce overall mortality?
Reviewer 2 Report
The authors have approached an interesting and relevant topic. Unfortunately, the study includes too many subgroup meta-analyses where most of the subgroups tested include only one study. Authors are encouraged to remove all subgroup meta-analyses focused on the cancer type and follow-up period, evaluate the suitability of the remaining data and, if appropriate, resubmit a manuscript.
Author Response
Reviewer 2
The authors have approached an interesting and relevant topic. Unfortunately, the study includes too many subgroup meta-analyses where most of the subgroups tested include only one study. Authors are encouraged to remove all subgroup meta-analyses focused on the cancer type and follow-up period, evaluate the suitability of the remaining data and, if appropriate, resubmit a manuscript.
- We thank the reviewer for taking time to review our manuscript and suggesting valuable suggestions to improve our manuscript.
Studies on cancer survival are relatively limited and thus, our meta-analysis is based on a limited number of studies. Yet, exploring heterogeneity in the relationship through subgroup analyses is a critical part of the meta-analysis and helps provide a complete picture of the evidence on the subject matter.
We do understand the reviewer’s concern and this is why we set the following rule in the method section of the original manuscript (“Of note, because results of a meta-analysis of a small number of studies are not informative, subgroup analyses were not conducted when the total number of studies was less than five.”). In fact, stratifying factors used in our subgroup analyses are critical to informing on gaps in the current evidence, and also informing future guidelines regarding vitamin D supplement use for cancer patients and thus, we kept them for future reference.
Round 2
Reviewer 2 Report
Inferences coming from a subgroup meta-analyses where most of the levels of the moderator variable include a single primary study are by definition misleading. The number of studies being more than five in subgroup meta-analyses does not prevent the misleading character of inferences if most of the levels include only one study.